# Safety Evaluation of *Weissella cibaria* JW15 by Phenotypic and Genotypic Property Analysis

**DOI:** 10.3390/microorganisms9122450

**Published:** 2021-11-27

**Authors:** Ye-Ji Jang, Hee-Min Gwon, Woo-Soo Jeong, Soo-Hwan Yeo, So-Young Kim

**Affiliations:** Fermented and Processed Food Science Division, Department of Agrofood Resources, National Institute of Agricultural Science, Rural Development Administration, Wanju 55365, Korea; rowlsla94@korea.kr (Y.-J.J.); vitamin89@korea.kr (H.-M.G.); wjddntnek@korea.kr (W.-S.J.); yeobio@korea.kr (S.-H.Y.)

**Keywords:** *Weissella cibaria*, safety, ARGs, virulence gene, toxic metabolite

## Abstract

*Weissella cibaria* is one of the bacteria in charge of the initial fermentation of kimchi and has beneficial effects such as immune-modulating, antagonistic, and antioxidant activities. In our study, we aimed to estimate the safety of *W. cibaria* JW15 for the use of probiotics according to international standards based on phenotypic (antibiotic resistance, hemolysis, and toxic metabolite production) and genotypic analysis (virulence genes including antibiotic resistance genes). The results of the safety assessment on *W. cibaria* JW15 were as follows; (1) antibiotic resistance genes (ARGs) (kanamycin and vancomycin etc.) were intrinsic characteristics; (2) There were no acquired virulence genes including Cytolysin (*cylA*), aggregation substance (*asa1*), Hyaluronidase (*hyl*), and Gelatinase (*gelE*); (3) this strain also lacked β-hemolysis and the production of toxic metabolites (D-lactate and bile salt deconjugation). Consequently, *W. cibaria* JW15 is expected to be applied as a functional food ingredient in the food market.

## 1. Introduction

In 1965, “probiotics” were first described as growth-promoting factors produced by microorganisms [1] and current probiotics are defined as ‘live micro-organisms which when administered in adequate amounts confer a health benefit on the host’ by the Food and Agriculture Organization of the United Nations and the World Health Organization (FAO/WHO) [2]. This characteristic of probiotics has been observed in bacteria, yeast, and fungi, but commonly used probiotics belong to lactic acid bacteria (LAB) and bifidobacteria, and their species are as following *Lactobacillus* (*L. acidophilus*, *L. gasseri*, *L. delbrueckii* subsp. *bulgaricus*, and *L. helveticus*), *Lacticaseibacillus* (*L. casei*, *L. paracasei*, and *L. rhamnosus*), *Limosilactobacillus* (*L. fermentum* and *L. reuteri*), *Lactiplantibacillus* (*L. plantarum*), *Ligilactobacillus* (*L. salivarius*), *Lactococcus* (*Lc. lactis*), *Enterococcus* (*E. faecium* and *E. faecalis*), *Streptococcus* (*S. thermophilus*), and *Bifidobacterium* (*B. bifidum*, *B. breve*, *B. longum*, and *B. animalis* subsp. *lactis*) [3,4].

Commercial starter culture products have been constantly consumed in the fermented food market as microbial food cultures (MFC) including LAB. Probiotic strains of LAB are also used in diverse medical and health-related areas, including the treatment of infections during pregnancy; management of allergic diseases; alleviation of intestinal inflammation; halt of antibiotic-related diarrhea, and prevention of urinary tract infections [5]. Of these, several probiotics such as *L. rhamnosus* GG (LGG), *B. animalis* subsp. *lactis* BB-12, and *L. casei* Shirota, etc. have been developed by global companies in the field of probiotics and they have also recently been marketed in the form of tablets or powders [6,7].

Probiotics are well-known and generally classified as safe (GRAS) because of their long-term safety in dairy products or fermented foods. The Lactic Acid Bacteria Industrial Platform (LABIP) has reported that the risk of infection caused by LAB occurs very rarely except for enterococci and streptococci [7]. However, in recent years, many controversies have been raised over the safety of probiotics since bacteria used in probiotics are frequently isolated from infection sources [3]. It should be noted that not all LAB of a particular genus or species have probiotic properties and are assigned to a particular strain such as *Lactobacillus delbrueckii* subsp. *bulgaricus* and *Streptococcus salivarius* subsp. *thermophilus* [8].

To evaluate the safety of probiotics, guidelines that take into account several factors in advance, such as excessive immune stimulation in sensitive individuals, systemic infection, gene transfer, or deleterious metabolic effects, are needed [9]. In 2002, FAO/WHO notes that it is important to conduct safety assessments, including production of certain metabolites such as D-lactate and ammonia, adverse effects in humans, antibiotic resistance, potential hemolysis, and toxin production, even for microorganisms classified as GRAS [2].

*Weissella* sp. is an LAB with the phenotypic properties of being Gram-positive, non-spore-forming, non-motile, etc. The genus *Weissella* is detected in various natural environments such as fermented foods (kimchi, fermented fava-bean, sausages etc.) and digestive tracts of humans and animals [10,11]. Many studies have shown that *W. cibaria*, which appears pro-dominant in the initial fermentation of kimchi, has beneficial effects such as probiotic properties, antimicrobial-, antagonistic-, and antioxidant-activities, etc. [12,13,14]. In addition to animal trials, the immunomodulatory activity of *W. cibaria* JW15 was significantly higher than that of *L. rhamnosus* GG, a well-known immune enhancer [12]. Recently, *W. cibaria* has been registered as a safe raw material by the Korea Food and Drug Administration (KFDA) [4] and is actively commercialized as a food ingredient in Korea. In 2018, the International Dairy Federation (IDF) is suggesting the use of *W. cibaria* as the Microbial Food Cultures (MFC) for the food usage of vegetables [15].

This study aimed to verify the safety of *W. cibaria* JW15 according to the international standards of FAO/WHO based on phenotypic (antibiotic resistance, hemolysis, and toxic metabolite production) and genotypic analysis (virulence genes including antibiotic resistance genes). Furthermore, *W. cibaria* JW15 was evaluated by the bacterial reverse mutation to identify genotoxicity.

## 2. Materials and Methods

### 2.1. Bacterial Strains and Growth Conditions

In our study, we used three *Weissella cibaria* strains for safety evaluation. Based on a previous study, *Weissella cibaria* JW15 (registered as KACC 91811P, Accession: NZ_CP-CP058237-CP058240), which is isolated from Kimchi, was selected as a bacterium that had immune-enhancing properties through NK cell activation [14]. *Lactobacillus rhamnosus* GG (ATCC 53103), *W. cibaria* LMG 21843, and *W. cibaria* LMG 17699 were purchased from the American Type Culture Collection (Manassas, VA, USA) and the BCCM/LMG Bacteria Collection (Ghent, Belgium) and used as reference strains. The *W. cibaria* strains and *L*. *rhamnosus* GG were cultivated anaerobically in De Man, Rogosa, and Sharpe broth (MRS broth, Merk, Darmstadt, Germany) at 30 °C for 24 h.

### 2.2. Minimum Inhibitory Concentrations on Antibiotics

The minimum inhibitory concentration (MIC) was determined using a commercial E-test (Epsilometer test, bioMerieux, France): Ampicillin, Gentamicin, Kanamycin, Streptomycin, Erythromycin, Clindamycin, Tetracycline, and Chloramphenicol. The concentration on the strips ranged from 0.016 to 256 μg/mL except for streptomycin (0.064 to 1024 μg/mL). Bacterial suspensions were adjusted to a turbidity of 0.5 of the McFarland standard (bioMerieux, Marcy l’Etoile, France). The suspensions were inoculated using a sterile cotton swab on the entire surface of the MRS agar plate. E-test strips were placed on the surface of the inoculated agar and anaerobically incubated at 35 °C for 48 h. The MIC was interpreted as the point at which the ellipse intersected the E-test strip according to the manufacturer’s instructions.

### 2.3. Detection of Antibiotic Resistance Genes

Chromosomal DNA was extracted from the bacteria using an AllPrep^®^ Bacterial DNA/RNA/Protein Kit (Qiagen, Hilden, Germany) according to the manufacturer’s instruction manuals. Plasmid DNA was also extracted from 2 mL of bacterial culture using an EzPureTM Plasmid prep kit (Enzynomics, Inc, Daejeon, ROK) according to the manufacturer’s instructions with the cell lysis buffer (20 mM Tris-Cl, pH 8.0, 2 mM sodium EDTA, and 1.2% Triton X-100 containing 20 mg/mL lysozyme). The quantity of DNA was assessed with a Nanodrop spectrophotometer (2000, Thermo Fisher Scientific, Waltham, MA, USA). The chromosomal and plasmid DNA extracts from the strains were diluted and used for polymerase chain reactions (PCRs) to detect ARGs targeted by gene-specific primers [16,17,18,19,20]. The PCR was performed with an initial denaturation step of 95 °C for 5 min, followed by 35 cycles of 95 °C for 10 s, annealing temperature (Table 1) for 30 s, 72 °C for 60 s. To confirm the amplicons of ARGs, PCR products were loaded into in agarose gels for electrophoresis. The primer sequences are listed in Table 1.

### 2.4. Genomic Analysis

We performed the genomic analysis of the JW15 strain using Macrogen service (Macrogen Inc., Seoul, Korea). The manufacturer’s instructions were as follows; DNA samples were sequenced using the PacBio RS II platform and Illumina HiseqXten platform, and then the subheads generated from PacBio RS II were assembled using the hierarchical genome assembly process (HGAP) [21] with default options. For error correction, the Illumina raw reads were filtered by quality at a level of 90% of bases had a phred score of 30 or higher. The assembly was corrected using high-quality HiseqXten reads by Pilon v1.21 [22]. Prokka v1.13 [23] was used for gene prediction and basic annotation. For additional annotation, the predicted protein sets were subjected to InterProScan v5.30-69.0 [24] and psiblast v2.4.0 [25] with EggNOG DB v4.5 [26]. Circular maps depicting each contig were generated using Circos v0.69.3 [27].

### 2.5. Bioinformatic Analysis of Virulence Factor-Related Genes

The VF-and toxin genes in the genome of the strain JW15, including Contig 1 to 4, were searched through BlastX analysis using Diamond software (ver. 0.9.26.127) [28] based on the virulence factor database (VFDB, http://www.mgc.ac.cn/VFs/, accessed on 12 August 2021) [29], which is an integrated and comprehensive online resource for curating information about virulence factors of bacterial pathogens. Thresholds for percent identity (% ID) and minimum length were set at 50% and 70%, respectively. In detail, VF-related genes, including those associated with enterotoxin, leukotoxin, cytolysin, cytotoxin K, hemolysis, biogenic amine production, hyaluronidase, aggregation, enterococcal surface protein, endocarditis antigen, collagen adhesion, cereulide, sex pheromone, and serine protease. These genes were additionally confirmed through BlastX analysis using experimentally-verified VF and toxin genes in the UniRef90 database.

### 2.6. Hemolytic Activity

*W. cibaria* strains, LGG, and *Bacillus cereus* KACC 10004 were used as positive controls for hemolytic activity. The strains were aerobically cultured in blood agar supplemented with 5% sheep blood at 37 °C for 2 days. The plates were then analyzed for microbial hemolytic properties by illuminating and observing the plate. Colonies that revealed green-hue zones (α-hemolysis) or did not reveal any hemolysis (γ-hemolysis) were considered non-hemolytic strains. Colonies that displayed blood lyses zones (clear zones) were classified as hemolytic (β-hemolysis) strains.

### 2.7. D-Lactic Acid Measurement

The production of D-lactic acid by *W. cibaria* strains and LGG was measured using the d-lactate colorimetric assay kit from BioVision Research (Mountain View, CA, USA). The LAB strains were cultured in MRS broth for 24 h at 37 °C, and the supernatant was used for this experiment.

### 2.8. Bile Salt Deconjugation

Bile salt deconjugation was carried out according to the plate assays of Dashkevicz and Feifhner [30]. *W. cibaria* strains and LGG were cultured for 24 h at 37 °C on an MRS agar plate containing 0.5% taurodeoxycholic acid (TDCA; Sigma, St. Louis, MO, USA). The results were interpreted as positive in the case of the formation of a halo of sediment or opaque granular white colonies around the colonies.

### 2.9. Enzymatic Profiles by API ZYM

Use of the API ZYM kit (bioMérieux, Marcy l’Etoile, France) was based on a substrate availability of a total of 19 enzymes. The bacterial suspension was adjusted with McFarland no. 5 being dropped in each tube. After incubation at 37 °C for 4 h, the results were determined to be positive if the color intensity was more than three following the manufacturer’s instructions.

### 2.10. Bacterial Reverse Mutation Assay

A bacterial reverse mutation assay was performed to evaluate the mutagenicity of *W. cibaria* JW15 with or without the S9 mix, following the principles of OECD Guideline 471 (2020) [31]. The assay was carried out using *Salmonella typhimurium* histidine-auxotrophic strains TA98, TA100, TA1535, TA1537, and *Escherichia coli* tryptophan-auxotrophic strain WP2uvrA (Molecular Toxicology, Boone, INC, USA). The S9 mix was used as a metabolic activation system (ORIENTAL YEAST Co., Ltd., Tokyo, Japan) and was prepared at the time of use in the required amount. Freeze-stored S9 (Lot No.: 20121110) and Cofactor A (Lot No.: A20120810) were thawed and prepared by mixing at a ratio of 1:9. Different dilutions of *W. cibaria* JW15 samples (5000, 2500, 1250, 625, and 313 μg/plate) were used for all tests under the same conditions. After being cultured at 37 °C for 48 h, the number of colonies in each tested group was counted per plate. This result was determined to be positive when the revertant colonies in the subject group were more than doubled. The data of historical control is presented in Appendix A.

## 3. Results and Discussion

The aim of this study was to verify the safety of *W. cibaria* JW15 based on phenotypic (antibiotic resistance, hemolysis, and toxic metabolite production) and genotypic analyses (virulence genes including antibiotic resistance genes). Currently, *W. cibaria* has no use as a probiotic ingredient, and the species is reported on antibiotic resistance such as kanamycin and vancomycin. Nevertheless, they have been frequently isolated from fermented foods and human feces and are well-known for their beneficial effects such as probiotic properties, antimicrobial-, antagonistic-, and antioxidant activities etc. Many researchers or consumers expect higher functional or healthy foods made from lactic acid bacteria with novel activity.

### 3.1. Determination of Minimum Inhibitory Concentrations

*Weissella* spp. has not been cleared on the cut-off values of MIC against antibiotics by EFSA in 2012. Accordingly, we determined an antibiotic susceptibility test of the *W. cibaria* JW15 strain corresponding to a *Leuconostoc* spp., based on the EFSA cut-off value, which reflects the phylogenetic and phenotypic characterization of the JW15 strain.

To ensure safety, the phenotypic antibiotic susceptibility of *W. cibaria* JW15 was investigated against 9 antibiotics, including ampicillin (AM), chloramphenicol (CL), clindamycin (CM), erythromycin (EM), gentamicin (GM), kanamycin (KM), streptomycin (SM), tetracycline (TC), and vancomycin (VA) using the E-test method [32]. As shown in Table 2, *W. cibaria* JW15 was susceptible to 7 kinds of antibiotics, including ampicillin (AM), chloramphenicol (CL), clindamycin (CM), erythromycin (EM), gentamicin (GM), streptomycin (SM), and tetracycline (TC), which were found below the cut-off value (μg/mL) within the safe range. However, the JW 15 strain was shown to be resistant to kanamycin (KM).

Recent studies have shown that the antibiotic susceptibility profile of *W. cibaria* differs between each strain [33,34]. *W. cibaria* CMU was found to be sensitive to AM, CL, CM, EM, GM, SM, and TC, except for KM corresponding to an obligate hetero-fermentative lactobacilli [33]. In our result, *W. cibaria* strains showed MICs ≥ 256 mg/L for kanamycin and vancomycin, suggesting that the resistance against kanamycin and vancomycin could be considered an intrinsic property. Antibiotic resistance was found not only in the genus *Weissella,* but also in many lactic acid bacteria used as food ingredients. *Lactobacillus* sp. shows high resistance to antibiotics reported as endogenous with strong resistance to antibiotics such as kanamycin and vancomycin [17,34]. It has been reported that lactic acid bacteria derived from fermented food are resistant to antibiotics [35,36]. Therefore, it has been speculated that the characteristic that the JW15 strain isolated from kimchi is resistant to some antibiotics may be common.

### 3.2. Detection of Antibiotic Resistance Genes

The transferability of antibiotic resistance (AR) genes and plasmids present in bacteria is associated with human health. Here, we confirmed the existence of AR genes and plasmids in the *W. cibaria* JW15 strain on four antibiotics (clindamycin, kanamycin, streptomycin, and vancomycin) showing high MIC cut-off value presented in Table 2.

PCR analysis for four antibiotic resistance genes such as streptomycin (*aad*A, *aad*E, and *str*B), tetracycline (*tet* (K)), kanamycin (*aph* (3”)-III and *ant* (2”)-I), and clindamycin (*Inu* (A) and *Inu* (B)) were conducted. Although there was detected amplicons in several samples, they were not antibiotic resistant genes based on sequencing analysis. The PCR results are shown in Table 3. There was no expected amplicon in the chromosome and plasmid DNA of *W. cibaria* JW15, *W. cibaria* LMG 21843, *W. cibaria* LMG 17699, and *L. rhamnosus* ATCC 53103 used in this study.

For antibiotic resistance, the MIC cut-off value of kanamycin was exceeded, which is a phenotypic evaluation, but the antibiotic resistance target gene was not detected in the chromosome and plasmid of the JW15 strain, which is a genotype evaluation. Sharma et al. (2014) reported that antibiotics intrinsic strains were phenotypically resistant may be genotypically susceptible [37]. We found several studies showing this characteristic, and strains that also had specific antibiotic resistance, but no gene was detected [33,38]. Therefore, the results of antibiotic resistance to KM and detection of their target genes are similar to those seen in antibiotic intrinsic strains according to previous reports. In addition, the phenotypic property of the JW15 strain that exhibits resistance to kanamycin may be due to four endogenous-related mechanisms such as enzyme inactivation or modification, alteration of bacterial target sites, antibiotic efflux pump and outer membrane permeability change, and intracellular metabolic rearrangement [37].

Moreover, for the transferability of antibiotic resistance, the plasmid plays a major role in the ARG gene transfer method (HGT) [37]. In our results, as shown in Table 3, kanamycin resistance gene (aph (3″)-III and ant (2″)-I) were not detected in the plasmid of JW15, thus the transferability is considered low.

### 3.3. Genomic Features of JW15 Strain

The key genomic features of *W. cibaria* JW15, including GC skew, protein-coding sequences (CDSs), COG categories, and G+C contents, are graphically depicted in Figure 1. The genome of the JW15 strain was a single circular chromosome of 2,472,214 bp with 3 plasmids (30,944 bp, 17,267 bp, and 14,411 bp). The genome of strain JW15 contains a total of 2315 CDS, 42 tRNA genes, and 28 rRNA genes. The result of COG-assigned proteins in the genomes of strain JW15 and their distributions into COG categories was not abbreviated. As a result, the COGs were classified into 26 functional categories except for Nohit against the COG database and of the 2556 protein-coding genes, 2259 genes (88.39%) were assigned to COGs categories. The *W. cibaria* UTNGt21O strain (1635 genes) reported by Tenea and Hurtado [39] was less than the *W. cibaria* JW15 strain. We found that the essential genes from the functional subcategories with the COG codes G (Carbohydrate transport and metabolism, 7.75%), J (Translation, ribosomal structure, and biogenesis, 7.67%), K (Transcription, 5.95%), L (Replication, recombination and repair, 4.38%), H (Coenzyme transport and metabolism, 3.44%), and I (Lipid transport and metabolism, 3.4%). However, the distribution of functional annotation of *W. cibaria* UTNGt21O strain was differently expressed in the order of R (General function prediction only, 8.99%), J (Translation, ribosomal structure, and biogenesis, 8.07%), K (Transcription, 6.60%), and L (Replication, recombination and repair, 6.54%) in comparison with *W. cibaria* JW15 strain. Owing to different genes, depending on strain-specificity, the information of functional genes mentioned here will help additional studies of this strain and demonstrate its potential property for the use of probiotics.

### 3.4. Bioinformatic Analysis of VF-Related Genes

In our study, we did not discover virulence-related genes in the chromosomal and plasmid genomes of the *W*. *cibaria* JW15 strain as shown in Table 4. However, two genes in JW15_contig1 (Table 5) were identified with low homology (loose identity, <95%) with the virulence factor database (VFDB) containing information on the virulence genes of bacterial pathogens online. Gene JW15-00598 showed a homology of 57.4% to gene efaA involved in endocarditis antigen, and gene JW15-00853 showed a homology of 53.1% to gene CD1208 (or CVF417) involved in RNA methyltransferase (or hemolysin A). In addition, the two genes (gene JW15-00598 and JW15-00853) researched in NCBI and Uniprot were found to have high homology (>95%) with transporter substrate-binding protein, RNA methyltransferase or cell division, respectively. In particular gene JW15-00853, which was identified as TlyA, is known to be not, on its own, a potent hemolysin [40]. Therefore, the two genes are presumed to be general transporters and transferase genes that are not related to toxic genes such as endocarditis antigen and hemolysin. In addition, the *W. cibaria* JW15 strain was negative in the hemolysis test, which was consistent with the bioinformatic analysis. The gene sequences are shown in Appendix A. The VF-related gene information that was used to confirm the safety of the strain should help further probiotic studies.

### 3.5. Toxic Metabolite Production

#### 3.5.1. Hemolytic Activity

Hemolysin is a toxic enzyme of pathogenic bacteria such as *Bacillus cereus* and has hemolytic activity to destroy red blood cells in the host, as well as the possibility for edema and anemia [38,41]. Generally, β-hemolysis is associated with microbial pathogenicity. In our study, *B. cereus* KACC 10004 as a positive control showed clear zones (expressed as β-hemolysis) around the colonies, whereas *W. cibaria* strains and LGG did not show β-hemolysis activity (Figure 2).

#### 3.5.2. D-Lactic Acid Production

Various bacterial species are known to produce D-lactate or both D- and L-lactates are produced in fermentation. Of them, the genus *Lactobacillus* produces D- and L-lactates, the genus *Pediococcus* produces L-, and the genera *Leuconostoc, Oenococcus,* and *Weissella* produce D-lactic acid [42]. As shown in Table 6, the productivity of D-lactic acid by *W. cibaria* was measured by enzymatic assays concerning D-lactate dehydrogenase. *W. cibaria* strains did not produce D-lactic acid like the commercial probiotic strain LGG. Our result was similar to the report showing that the *W. cibaria* CMU strain was unable to produce D-lactic acid [33].

#### 3.5.3. Bile Salt Deconjugation Test

Bile salts are less capable of solubilizing and absorbing lipids in the gut. All strains used in this study were able to grow in the presence or absence of sodium taurodeoxycholate (0.5%) and did not show the precipitate halos or the opaque white colonies after growth in MRS with TDCA (Table 6). These results show the lack of ability to deconjugate sodium taurodeoxycholate and agree that *W. cibaria* could not convert to secondary bile acid as previously published report [36].

#### 3.5.4. Enzymatic Profile by API ZYM

The enzyme profile of the JW15 strain was similar to that of the LMG 28143 strain isolated from fermented kimchi, while the LMG 17699 strain was different from the β-galactosidase β-glucosidase enzymes (Table 6). In Muñoz-Atienza et al. (2013), it was observed that leucine arylamidase, valine arylamidase, β-galactosidase, and β-glucosidase showed different patterns among the 15 kinds of *Weissella* spp. [36]. In the case of the β-glucuronidase, no generation of potential carcinogenic metabolites [43] was detected in any of the *W. cibaria* strains and LGG, indicating that they are safe.

### 3.6. Bacterial Reverse Mutation Assay

The bacterial reverse mutation assay, developed by Bruce Ames in 1973 [44], was performed for mutagenicity testing of probiotics such as *L. rhamnosus*, *B. adolescentis*, *L. paracasei*, *L. mali*, and *P. acidilactici* [45,46,47]. The genotoxicity was conducted by this assay with different doses of *W. cibaria* JW15 against four mutant *S. typhimurium* strains (TA98, TA100, TA1535, and TA1537) and a mutant *E. coli* strain (WP2uvrA), respectively. An expected increase of revertant colonies was observed in all positive groups after induction of the mutants. After exposure of bacterial strain to different concentrations of *W. cibaria* JW15, the number of revertant colonies, regardless of the presence or absence of S9 mix, did not exceed twice that of the negative control group (Table 7). Therefore, the *W. cibaria* JW15 treatment groups were considered not to have mutagenic activity in the histidine auxotrophy of the *S. typhimurium* strains or the tryptophan auxotrophy of *E. coli*.

Consequently, in this study, we verified the safety of the *W. cibaria* JW15 strain by phenotypic and genotypic property analysis according to the international guidelines by FAO/WHO. The safety was evaluated by a minimum inhibitory concentration assay for 9 antibiotics, chromosomal and plasmid DNA analysis for 12 antibiotic resistance genes (ARGs) on 4 antibiotics, virulence gene analysis, beta-hemolysis, toxic metabolite production, and bacterial reverse mutation assay. The strain *W. cibaria* JW15 was susceptible to all antibiotics except for kanamycin and vancomycin. We confirmed that there was no harboring of antibiotic resistance target genes and virulence-related genes in the genome of strain JW15. We therefore considered that antibiotic resistance (e.g., kanamycin, vancomycin) was an intrinsic property of *W. cibaria* JW15. Additionally, the strain JW15 lacked β-hemolysis, β-glucuronidase, toxic metabolites such as D-lactate and bile salt deconjugation, and bacterial reverse mutagenic activity. Accordingly, we believe that *W. cibaria* JW15 could be commercially applied as a probiotic strain in the future.

## Figures and Tables

**Figure 1 microorganisms-09-02450-f001:**
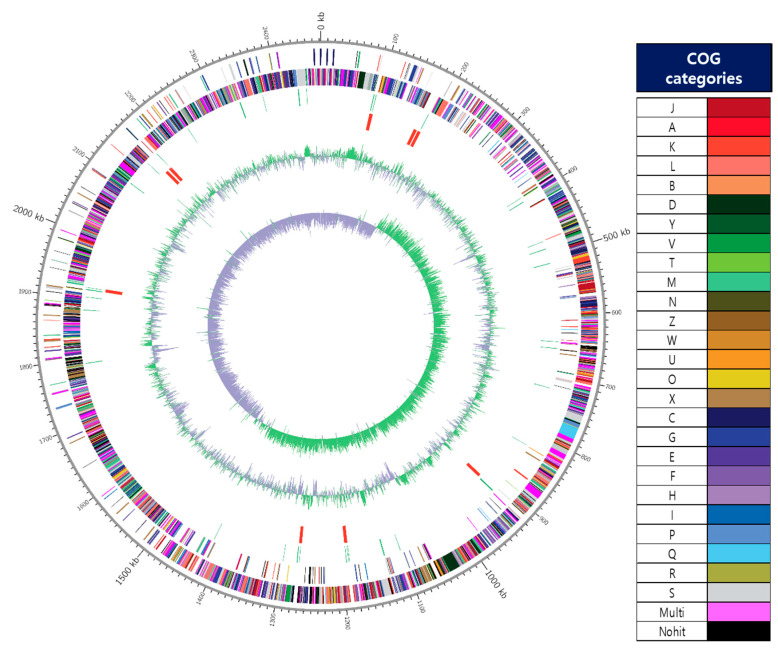
A circular map of the chromosome of JW15 strain. The map was drawn by applying Contig 1’s annotation result. From outside to the center: coding sequences (CDS) on forward strand (colored by COG categories of the right side), CDS on reverse strand (colored by COG categories of the right side.), tRNA, rRNA, GC content, and GC skew (+: green, −: violet). The complete genome contained 2,472,214 bp with G+C content of 45.09%.

**Figure 2 microorganisms-09-02450-f002:**
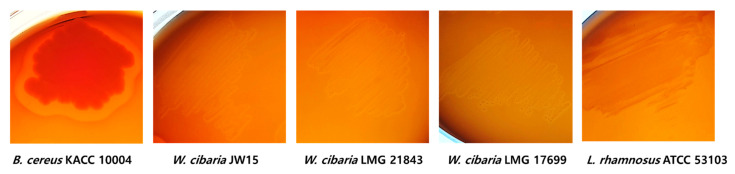
Hemolytic activity of *W. cibaria* strains and LGG. Complete lysis of blood cells was observed, with clear zones around *B. cereus* KACC 10004 as positive control.

**Table 1 microorganisms-09-02450-t001:** Primer and gene sequences used in this study.

Anti Biotics	Target Genes	Sequences (5′-3′)		Tm (°C)	Amplicon Size (bp)
		F	R		
Streptomycin	*aad*A	ATCCTTCGGCGCGATTTTG	GCAGCGCAATGACATTCTTG	53	282
	*aad*E	ATGGAATTATTCCCACCTGA	TCAAAACCCCTATTAAAGCC	53	565
	*str*B	ATCGTCAAGGGATTGAAACC	GGATCGTAGAACATATTGGC	55	509
Tetracycline	*tet*(K)-1	TCGATAGGAACAGCAGTA	CAGCAGATCCTACTCCTT	51	169
	*tet*(K)-2	TTATGGTGGTTGTAGCTAGAAA	AAAGGGTTAGAAACTCTTGAAA	52	348
	*tet*(K)-3	TTAGGTGAAGGGTTAGGTCC	GCAAACTCATTCCAGAAGCA	55	697
Kanamycin	*aph*(3″)-III	GCCGATGTGGATTGCGAAAA	GCTTGATCCCCAGTAAGTCA	57	292
	*ant*(2″)-I	GGGCGCGTCATGGAGGAGTT	TATCGCGACCTGAAAGCGGC	61	329
Clindamycin	*lnu*(A)	GGTGGCTGGGGGGTAGATGTATTAACTGG	GCTTCTTTTGAAATACATGGTATTTTTCGATC	59	323
	*Inu*(B)	CCTACCTATTGTTTGTGGAA	ATAACGTTACTCTCCTATTTC	52	925
**Gene Information**					
JW15-1_1_00598		MLKKLGLTAGALAIAIGGTVWFVQNRDAQTATASGELRVVTTNSILEDMVEQVGGDDVSVYSIVKRGTDPHEYEPKTA DITATTEANVIFHNGLNLETGGNGWFSKLTKTANKRDNQEVFSASRLVEPLFLTSKGKEDEMDPHAWLDLNNGIKYV KTITNVLKDKDPEHAQAFQKRSDAYIAKLRALHNEAKDKFADVPVEKRLLVTSEGAFKYFSKVYGIQPAFIWEINTESQ GTPEQMKQVLAKIAASNVKSLFVESSVSPKSMEKVSKETGLPIYEKIYTDSLAKKGTTWDTYYDMV			
JW15-1_1_00853		MAIEKERVDVLAVQQGLFTSREQAKRAIMAGEILGENEQRMDKAGEKIPVTTELHLKGAPMPYVSRGGFKLEKALEVF DISVQDKVVLDIGSSTGGFTDVSLQNGAKLVYALDVGTNQLVWKLRSDERVVVMENTNFRYSEPTDFTHGQPAVATID VSFISLNLILPPLAKILTPGGSVATLIKPQFEAGREAIGKHGIVKDATTHLAVLDKVAGYAQAAGFSIVALDYSPIKGGSG NIEFLAHLVLDGGESTMTEAEREAVVTRAHAQLNVRREENADETK			

**Table 2 microorganisms-09-02450-t002:** Antibiotic resistance profiles and minimum inhibitory concentration (MIC) values of bacterial strains used in this study.

Strains	Microbiological Cut-Off Values (mg/L) of Antibiotics									Ref.
	AM	CL	CM	EM	GM	KM	SM	TC	VA	
***Leuconostoc* spp.**	2	4	1	1	16	16	64	8	N/R	(EFSA ^a^, 2012)
***W. cibaria* JW15**	0.100 ± 0.000	2.300 ± 1.100	0.040 ± 0.030	0.700 ± 0.400	4.500 ± 2.100	>256	56.000 ± 11.300	1.800 ± 0.400	>256	In this study
***W. cibaria* LMG 21843**	0.056 ± 0.008	2.000 ± 0.000	1.000 ± 0.000	0.750 ± 0.250	6.000 ± 2.000	>256	96.000 ± 0.000	1.750 ± 0.250	>256	
***W. cibaria* LMG 17699**	0.095 ± 0.031	1.750 ± 0.250	0.044 ± 0.021	0.875 ± 0.125	20.000 ± 4.000	>256	96.000 ± 0.000	1.750 ± 0.250	>256	
***L. rhamnosus* spp.**	4	4	1	1	16	64	32	8	N/R	(EFSA, 2012)
***L. rharnnosus* GG**	0.285 ± 0.095	1.250 ± 0.250	0.315 ± 0.185	0.235 ± 0.145	96.000 ± 0.000	>256	96.000 ± 0.000	0.470 ± 0.280	>256	In this study

^a^ European Food Safety Authority (EFSA) 2012; AM, ampicillin; CL, chloramphenicol; CM, clindamycin; EM, erythromycin; GM, gentamicin; KM, kanamycin; SM, streptomycin; TC, tetracycline; VA, vancomycin; N/R, not required.

**Table 3 microorganisms-09-02450-t003:** Detection for antibiotics resistance genes (ARGs).

Antibiotic	Target Gene	Ref.	*W. cibaria* JW15		*W. cibaria* LMG 21843		*W. cibaria* LMG 17699		*L. rharmnosus* ATCC 53103	
			Chromosome	Plasmid	Chromosome	Plasmid	Chromosome	Plasmid	Chromosome	Plasmid
SM	*aad*A	[16]	−	−	−	−	−	−	−	−
	*aad*E	[20]	−	−	−	−	−	−	−	−
	*str*B	[16]	−	−	−	−	−	−	−	−
TC	*tet*(K)-1	[19]	−	−	−	−	−	−	−	−
	*tet*(K)-2	[17]	−	−	−	−	−	−	−	−
	*tet*(K)-3	[20]	−	−	−	−	−	−	−	−
KM	*aph*(3″)-III	[16]	−	−	−	−	−	−	−	−
	*ant*(2″)-I	[16]	−	−	−	−	−	−	−	−
CM	*lnu*(A)	[17]	−	−	−	−	−	−	−	−
	*Inu*(B)	[18]	−	−	−	−	−	−	−	−

−: not detected; SM, streptomycin; TC, tetracycline; KM, kanamycin; CM, clindamycin.

**Table 4 microorganisms-09-02450-t004:** Bioinformatic analysis for the presence of putative virulence factor-related genes in the genomes of strain JW15.

Class	Gene	*W. cibaria* JW15	
		Contig 1	Contig 2, 3, 4
Enterotoxin	*selk, selq, set*	−	−
Leucotoxin	*lukD*	−	−
Cytolysin	*cylA*	−	−
Cytotoxin K	*cytK*	−	−
Hemolysin	*hbl*	−	−
Gelatinase	*gelE*	−	−
Amino acid decarboxylase	*hdc1, hdc2*	−	−
	*tdc*	−	−
	*odc*	−	−
	*ldc*	−	−
Hyaluronidase	*hyl*	−	−
Aggregation substance	*asa1*	−	−
Enterococcal surface protein	*esp*	−	−
Endocarditis antigen	*efaA*	−	−
Adhesion of collagen	*ace*	−	−
Cereulide	*cesA*	−	−
Sex pheromones	*ccf, cob, cpd*	−	−
Serine protease	*sprE*	−	−
Transposon-related genes	*int, intTN*	−	−
BLASTX results against VFDB was filtered based on followed thresholds;			
Coverage 70%			
Percent identity 50%			

**Table 5 microorganisms-09-02450-t005:** Homology analysis of two genes presumed to be VF-genes in the genome of *W. cibaria* JW 15 strain.

Locus Tag	VFDB Reference		NCBI-Nr Database Reference		Uniprot Database Reference	
	Id(%)	Gene Name	Id(%)	Gene Name	Id(%)	Gene Name
JW15-1_1_00598	57.4	(efaA) endocarditis specific antigen	100	zinc ABC transporter substrate-binding protein [*Weissella cibaria*]	98.7	SsaB protein (*Weissella cibaria*) (unreviewed)
					51.1	Metal ABC transporter substrate-binding lipoprotein (*Streptococcus pyogenes serotype*) (reviewed)
JW15-1_1_00853	53.1	putative RNA methyltransferase [Hemolysin (CVF417)]	100	TlyA family RNA methyltransferase [*Weissella cibaria*]	99.6	Cell division protein FtsJ (*Weissella cibaria*) (unreviewed)
					86.4	23S rRNA (Cytidine1920-2′-O)/16S rRNA (Cytidine1409-2′-O)-methyltransferase (*Weissella soli*) (unreviewed)

**Table 6 microorganisms-09-02450-t006:** Enzymatic profiles and assay of toxic metabolic production.

Enzymatic Profiles	JW15	LGG	LMG 21843	LMG 17699
Alkaline phosphatase	−	−	−	−
Esterase (C4)	−	+	−	−
Esterase lipase (C8)	−	+	−	−
Lipase (C14)	−	−	−	−
Leucine arylamidase	+	+	+	+
Valine arylamidase	−	+	−	−
Cystine arylamidase	−	+	−	−
Trypsin	−	−	−	−
α-chymotrypsin	−	+	−	−
Acid phosphatase	+	+	+	+
Naphthol-AS-BI- phosphohydrolase	+	+	+	+
α-galactosidase	−	−	−	−
β-galactosidase	−	+	−	+
β-glucuronidase	−	−	−	−
α-glucosidase	−	+	−	−
β-glucosidase	−	+	−	+
acetyl glucosaminidase	−	−	−	−
α-mannosidase	−	−	−	−
α-fucosidase	−	+	−	−
**Toxic metabolic production**				
Hemolysis (beta-)	−	−	−	−
D-lactate (nmol/μL)	0.010 ± 0.006	0.000 ± 0.000	0.019 ± 0.011	0.015 ± 0.007
Bile salt deconjugation	−	−	−	−

+: positive, −: negative.

**Table 7 microorganisms-09-02450-t007:** Mutagenic activity in bacterial strains TA98, TA100, TA1535, TA1537, and WP2uvrA treated with *W. cibaria* JW15, with (+S9) or without (−S9) metabolic activation.

Dose of *W. cibaria* JW15 (μg/Plate)	Number of Revertant Colonies per Plate (Mean ± S.D)				
	TA98	TA100	TA1535	TA1537	WP2uvrA
**Without S9 mix**					
Negative control ^a^	20.7 ± 0.9	103.7 ± 4.2	14.8 ± 0.7	8.7 ± 0.5	41.2 ± 12.5
313	22.3 ± 0.5	106.7 ± 6.1	13.3 ± 1.4	8.7 ± 2.4	42.7 ± 12.3
625	21.8 ± 0.2	102.3 ± 2.4	15.3 ± 0.5	8.3 ± 1.4	43.5 ± 12.5
1250	19.5 ± 1.6	106.3 ± 5.2	13.8 ± 1.6	9.5 ± 2.6	41 ± 13.2
2500	21.3 ± 1.4	101.7 ± 3.3	15.5 ± 0.2	9.7 ± 0.9	40.7 ± 14.1
5000	18.8 ± 1.2	105.3 ± 7.5	13.8 ± 0.2	8.7 ± 1.4	40.2 ± 12.5
Positive control ^b^	566.5 ± 18.1	621.2 ± 35.6	451.5 ± 0.7	440.5 ± 18.1	227.5 ± 5.9
**With S9 mix**					
Negative control ^a^	32 ± 0.5	112.5 ± 10.6	14.2 ± 0.2	17 ± 4.2	36.3 ± 5.2
313	31 ± 0.5	114.7 ± 10.4	13.3 ± 0	16 ± 4.7	35.7 ± 7.1
625	31.7 ± 1.9	113.5 ± 13.4	14 ± 0.9	16.2 ± 5.4	36.5 ± 5.9
1250	30.2 ± 0.2	111.3 ± 9.4	15.5 ± 0.2	17 ± 6.6	34.8 ± 6.8
2500	32.3 ± 0.9	114 ± 10.8	15.5 ± 2.1	16 ± 6.1	36.7 ± 5.7
5000	32.7 ± 0.5	110.3 ± 9	13.7 ± 0	17.3 ± 6.6	34.3 ± 6.6
Positive control ^c^	333.7 ± 46.2	667.8 ± 0.2	149.3 ± 14.6	158.8 ± 17.2	454.3 ± 31.1

^a^ Nomal saline injection; ^b^ positive control without S9 for TA98: 2-Nitrofluorene (2-NF), 5.0 μg/plate; for TA100 and TA1535: Sodiumazide (SA), 1.5μg/plate; for TA1537: 9-Aminoacridine (9-AA), 80.0 μg/plate; for WP2uvrA: 4-Nitroquinoline N-oxide (4-NQO), 0.3 μg/plate; ^c^ Positive control with S9: 2-Aminoanthracene (2-AA) for TA98, 1.0 μg/plate; TA100, 2.0 μg/plate; TA1535 and TA1537, 3.0 μg/plate; WP2uvrA, 10.0 μg/plate; S.D.: standard deviation.

## Data Availability

This genome sequence was deposited in GenBank (BioProject number PRJNA639573; and GenBank accession numbers CP058237-CP058240). The version described in this paper is the first version.

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
