# Peer review of "Safety Evaluation of Weissella cibaria JW15 by Phenotypic and Genotypic Property Analysis"

_microorganisms, 2021, doi:10.3390/microorganisms9122450_

Round 1

Reviewer 1 Report

This study provides a safety assessment of the probiotic strain Weissella cibaria JW15 based on phenotypic and genotypic analysis. The study reveals important new scientific information that supports the applicability of this strain in food products. It is well justified with a variety of methods that prove the safety of the bacterium and therefore the findings of this study have a significant commercial interest.

Some suggestions for improvement:

Line 9: “has been published the results of its” change to “has beneficial…”

Line 13: please provide explanation for ARGs

Lines 22: “described the growth” change to “described as the growth”

Line 23: “and defined a probiotic” change to “and a prebiotic is defined as..”

Line 38: please provide explanations for LGG, BB12

Lines 43-44: Change to “However, in recent years, many controversies have been raised over the safety of probiotics since bacteria used in probiotics have been frequently isolated from infection sources [3].”

Line 46: please clarify “are assigned to a particular strain”

Lines 58-59: change to “Many studies have shown that especially, W. cibaria, which appears pro-dominant in the initial fermentation of kimchi, has beneficial effects such as probiotic properties, antimicrobial-, antagonistic-, and antioxidant-activities etc.”

Lines 70-71: change to “Furthermore, W. cibaria JW15 was evaluated by the bacterial reverse mutation to identify the genotoxicity”.

Line 75: the specific strain was isolated in-house? Or was purchased?

Line 144: MRS broth?

Lines 179-182: this paragraph needs some more explanation, especially because it is at the beginning of the Results and Discussion.

Line 189: found instead of founded

Line 190: change to “was shown to be resistant to two kinds…”

Line 218:  change to “antibiotic resistance target genes”

Line 220-21: needs better explanation

Line 226: “are to be due” change to “may be due”

Line 256: please explain VFDB

Line 272: change to “complete genome contained”

Lines 280-282: change to “as positive control showed clear zones (expressed as β-hemolysis) around the colonies, whereas W. cibaria strains and LGG did not show the β-hemolysis activity (Figure 2).”

Table 3: please provide headings to the two last columns (eg number of protein-coding sequences ?)

Table 5: Make sure that all names of bacteria are in Italics

Lines 315-316: change to “These results show the lack of ability to deconjugate sodium taurodeoxycholate”

Line 343: change to “were considered not to have mutagenic…”

Table 7: these are the results of the positive and the negative controls, but where are the results of the W.cibaria JW15?

Conclusion: It would be nice to add a small paragraph as conclusion to summarize all the important findings of the study and address where further research should be done.

Author Response

We would like to thank the reviewer for the comments and suggestions made about this manuscript. We revised the manuscript according to the suggestion. In order to address the comments clearly, we illustrated the answers for the reviewer’s comments in detail in the manuscript.

Reviewer 2 Report

Major concern:

1. The authors have taken many sentences from the following paper: https://doi.org/10.3390/ijms20112693 

For example Lines 277-280. They should write their own sentences.

2. Hemolytic activity tested between anaerobic and aerobic should not be compared, comparison should be made from similar experimental conditions.

3. What are the references in Table 2 stand for? and what does '-' mean?

4. Hemolysis results are not clear. The photographs are unacceptable and difficult to interpret the results.

5. The color of the first row of Table 3 doesn't allow to read the column headings.

6. The sequences of genes at the end of Table 5 should be presented in some other way or be shown in the supplementary information.

7. There is no conclusion of the study provided.

Minor comments:

1. Line 124-128: incomplete sentence.

2. Line 259: strange sentence.

3. Line 188-190: Grammatical errors.

Author Response

We thank the reviewer for the comments and the suggestions put forth for clarifying this manuscript. We revised the manuscript according to the major and minor concern.

Round 2

Reviewer 2 Report

The authors have improved the manuscript based on the comments. I could not find the gene sequence in supplementary information and there is no Table S1. Did I miss it or was it missed by the authors? 

Author Response

We would like to thank the comments and re-uploaded the supplementary information. The sequence information can be found in Table S1.
